# Dynamics of Microorganisms and Metabolites in the Mixed Silage of Oats and Vetch in Alpine Pastures, and Their Regulatory Mechanisms Under Low Temperatures

**DOI:** 10.3390/microorganisms13071535

**Published:** 2025-06-30

**Authors:** Shuangpeng Xu, Guoli Yin, Xiaojun Yu

**Affiliations:** College of Pratacultural Science, Gansu Agricultural University, Lanzhou 730070, China; xsp0704@163.com (S.X.); yuxj@gsau.edu.cn (X.Y.)

**Keywords:** oats, vetch, mixed silage, microorganisms, metabolites

## Abstract

Silage is an effective method for alleviating winter feed shortages, but the mechanisms by which the silage microorganisms and metabolites respond to a mixture of oats and vetch at low temperatures remain unclear. In this study, the quality, microorganisms, and metabolites of oats mixed with vetch as a silage material, as well as after 90 days of silage, were analyzed. The traditional view holds that a decrease in microorganism diversity during silage indicates successful fermentation. However, in the present study, microorganism diversity was found to increase after silage under alpine and low-temperature conditions, with a significant rise in the abundance of microorganisms such as *Levilactobacillus* and *Kazachstania*. This phenomenon may be explained by the inhibition of rapid lactic acid bacteria proliferation by low temperatures, which allows for the survival of other cold-tolerant microorganisms and their involvement in metabolism. These microorganisms significantly increased the levels of metabolites such as l-methionine, l-glutamine, arachidonic acid, and linolenic acid in the mixed feeds, while simultaneously significantly decreasing the levels of metabolites such as l-leucine, l-arginine, l-asparagine, and glyceric acid. These metabolites possess antioxidant and anti-inflammatory properties that enhance the nutritional value of the feed and indirectly improve the immunity and performance of ruminants. This study comprehensively revealed the complex network of interactions between microorganisms and metabolites in the mixed forage of oats and vetch in alpine pastures and elucidated the regulatory mechanism of silage under low temperatures. The subsequent development of microorganism preparations for the targeted regulation of silage quality provides a theoretical foundation for producing high-quality silage in alpine pastures.

## 1. Introduction

China’s alpine pastoral areas are primarily located on the Tibetan Plateau, the Inner Mongolian Plateau, the northeastern high latitudes, and other cold-climate zones [1,2]. These regions are primarily focused on livestock production, with the farming of herbivorous animals such as sheep, yaks, and goats playing an important role in the local economy. However, due to the cold climate, natural pasture resources are often insufficient in winter and spring in these areas, severely limiting the development of animal husbandry [3]. According to statistics, the shortage of winter feed in alpine pastures can exceed 40%, forcing herders to rely on purchased feed and significantly increasing breeding costs. Therefore, how to process and store both ‘food and feed’ under alpine conditions, to improve the yield and nutritional value of feed, has become one of the key issues in modern grassland animal husbandry research. Oats and vetch are the most common forage crops in China’s alpine pastures. Oats are hardy, drought-resistant, cold-tolerant, and produce a high grass yield, accounting for about 70 percent of the area of artificially planted grass in the Tibetan Plateau region. Vetch is a high-quality legume forage, and mixing it with oats can significantly increase yield and improve forage quality [4,5].

Oats and vetch are typically sown in late spring and harvested in autumn to make hay for ruminants [4]. However, in alpine pastures, frequent light rains in late summer and autumn affect the production of high-quality hay. Silage, as a storage technology for the long-term preservation of feed, offers several advantages, including reduced nutrient loss, improved utilization value, regulation of the supply period, and enhanced digestibility and palatability for ruminants. It is an effective method for alleviating feed shortages in spring and winter in alpine pastures [6]. Mixed silage of oats and vetch is rich in nutrients. The amylose in oats is utilized by lactic acid bacteria to produce lactic acid, which rapidly lowers the pH, while the proteins in vetch are broken down by specific microorganisms to produce ammonia and other volatile compounds [7]. However, in alpine pastures, where the average daily temperature during the forage harvesting period is around 10 °C, the optimal temperature for silage is generally between 20 °C and 30 °C, and low temperatures inhibit the silage fermentation process. The quality of low-temperature silage fermentation is influenced by the microorganisms’ structure and metabolites [4]. The increase in harmful bacteria and the depletion of beneficial bacteria during the silage process can easily lead to the failure of silage fermentation. Previous studies have emphasized the role of bacterial communities in silage fermentation [8]. However, Anumudu found that yeasts and spoilage fungi consume water-soluble carbohydrates and lactic acid, produce mycotoxins, and raise the pH, thereby affecting silage fermentation quality [9]. The fermentation activity of microorganisms during the silage process not only changes the chemical composition of the feed but also produces a large number of metabolites. These metabolites not only affect the palatability of the feed but are also directly related to its preservation quality and nutritional value [10]. In oat and vetch mixtures, the silage process produces a richer array of metabolites due to the complementary nutrient content of the two [4,11].

Therefore, this study utilized oats and vetch to make silage at a natural temperature. The fermentation parameters, chemical composition, microorganisms, and metabolites were determined for the mixed-silage fresh material and fermentation over 90 days. The microorganisms and metabolites were also analyzed jointly. This study aimed to elucidate the composition of microorganisms and metabolites in fresh oat and vetch mixed forage, as well as the mechanism by which low temperature regulates microorganisms and metabolites in mixed silage, providing theoretical references for the production of high-quality oat and vetch mixed silage in alpine pastures.

## 2. Materials and Methods

### 2.1. Silage Preparation

In April 2023, the research team conducted field trials in the Gannan Tibetan Autonomous Prefecture, Gansu Province. The average altitude of the region is 2737 m, and the average annual temperature is 3.0 °C. Longyan No. 3 (*Avena sativa*) was used for oats, and Ximu 333 (*Lathyrus sativus* ) was used for vetch. These two varieties are widely distributed across the Tibetan Plateau and have similar maturation periods. Planting was carried out by sowing at 255 kg/hm^2^ and 60 kg/hm^2^, respectively. At the end of August of the same year, during the milky ripening period of oats and the flowering period of vetch, the forage was harvested with a disc mower at approximately 15 cm above the ground and left to dry under natural conditions. The moisture content of the forage was monitored using a moisture meter, and when it reached 65–70%, the forage was cut into small segments of about 1 cm using a kneading machine. Samples were collected to determine their initial nutrient composition, microbial communities, and metabolites. The chopped feed mix was evenly packed into polyethylene vacuum bags (45 cm × 75 cm, Suqian City Trading Co., Ltd., Suqian, China), with 500 g in each bag. The bags were then vacuum-sealed using a vacuum machine. A total of six silage bags were prepared and stored under a local hay storage shed for fermentation at natural temperature. After 90 days of storage, the silage bags were opened, and samples were collected for the determination of nutrient composition, fermentation quality, microorganisms, and metabolites.

### 2.2. Determination of Fermentation Parameters and Nutrient Content

A 20 g silage sample was homogenized with 180 mL of distilled water and macerated at 4 °C for 24 h. After extraction, the filtrate was filtered through sterile 4-layer gauze and collected for the determination of the fermentation parameters. The pH was measured using a glass electrode pH meter, and ammonia nitrogen (AN) was quantified using the phenol–hypochlorite colorimetric method [12]. Lactic acid (LA), acetic acid (AA), propionic acid (PA), and butyric acid (BA) were quantified using high-performance liquid chromatography (HPLC) [13]. The remaining samples were dried at 65 °C for 48 h to a constant weight to determine the dry matter (DM) content. The dried samples were crushed and ground for chemical composition analysis. Soluble carbohydrate (WSC), ether extract (EE), and crude protein (CP) content were determined according to the procedures of the Association of Official Analytical Chemists [14]. Neutral detergent fiber (NDF) and acid detergent fiber (ADF) were determined according to the method described by [15]. Crude ash (Ash) was determined using the muffle combustion method [16].

### 2.3. Microbial Community Analysis

Total microbiome DNA was extracted from fresh raw materials of an oat and vetch mixture, as well as silage samples, using the TGuide S96 Magnetic Bead Method Soil/Faecal DNA Extraction Kit (Tiangen Biochemical Science and Technology Co. Ltd., Beijing, China). After DNA isolation, nucleic acids were assessed for concentration and purity using an enzyme labeler (GeneCompang Limited, model Synergy HTX). The 16S rRNA V3-V4 region of genomic DNA was amplified using TaKaRa PrimeSTAR DNA polymerase (model DR500A). Bacterial primers were designed with reference to Liu using 338F (5′-ACTCCTACGGGGAGGCAGCA-3′) and 806R (5′-GACTACHVGGGGTATCTAATCC-3′); fungal primers were designed with reference to Scibetta using ITS1F (5’-CTTGGTCATTTAGAGGAAGTAA-3′) and ITS2R (5′-TCCTCCGCTTATTGATATATGC-3′) [17,18]. The amplification procedure was as follows: pre-denaturation at 95 °C for 3 min, followed by denaturation at 95 °C for 30 s, annealing at 55 °C for 30 s, extension at 72 °C for 30 s, and 40 cycles. The amplified PCR products were subjected to electrophoretic analysis to assess the integrity using an agarose gel at a 1.8% concentration (Beijing Bomei Fuxin Science and Technology Co., Beijing, China). For the constructed libraries, the mixed products underwent damage repair, end repair, and ligation of junctions using the SMRTbell Template Prep Kit provided by PacBio. The reaction was performed on a PCR instrument, and the final library was purified and recovered using AMPure PB magnetic beads. The final libraries were quantified using Qubit for the concentration and assessed for size using the Agilent 2100 to confirm compliance with the onboarding requirements (Invitrogen, Thermo Fisher Scientific, Hillsboro, OR, USA). Sequencing was performed on a Sequel II sequencer (Beijing Biomarker Technologies Co., Ltd., Beijing, China). After sequencing, the data were processed using QIIME (version 1.8.0) software. Low-quality sequences and nonspecific sequences (including adapters, chimeras, poly-A, and primers), as defined by Gill and Chen, were filtered using Trimmomatic (version 0.33) [19,20]. Valid sequences were clustered at 97% similarity using USEARCH (version 10.0) software, and species were annotated using UCHIME (version 8.1). Species annotation was performed through the Silva database (Release 138, http://www.arbsilva.de) to analyze the microorganisms’ structure and species composition of each sample. The alpha diversity index was calculated using QIIME2 software (version 1.7.0), and dilution curves and rank abundance curves were plotted for the samples. Beta diversity analysis was performed to assess changes in the composition and structure of the sample flora. Heatmap clustering and principal coordinate analysis (PCoA) were conducted on the R language platform. Differential biomarkers between treatments were analyzed using linear discriminant analysis (LDA) effect size (LefSe). Species abundance data were compared between groups using *t*-tests in Metastats2 software (version 1.0) to identify significantly different species between samples from different treatments.

### 2.4. Metabolic Profile Analyses

Weigh 50 mg of the sample, add 1000 μL of the extraction solution containing the internal standard (methanol–acetonitrile–water = 2:2:1, internal standard concentration 20 mg/L), and vortex for 30 s. Steel beads were then added, and the sample was processed for 10 min in a 45 Hz grinder, followed by sonication for 10 min in an ice-water bath. The samples were allowed to stand at −20 °C for one hour, then centrifuged at 4 °C and 12,000 rpm for 15 min. Carefully transfer 500 μL of the supernatant to an EP tube and dry the extract in a vacuum concentrator. Add 160 μL of extract (acetonitrile–water = 1:1) to the dried metabolites. After repeating the vortexing, sonication, and centrifugation, carefully transfer 120 μL of the supernatant to a 2 mL injection vial. Ten μL from each sample was taken and mixed into a QC sample for online testing. The LC/MS system for metabolomics analysis is composed of a Waters Acquity I-Class PLUS ultra-high performance liquid tandem Waters Xevo G2-XS QTof high resolution mass spectrometer (Milford, MA, USA). The column used is purchased from Waters Acquity UPLC HSS T3 column (1.8 μm 2.1 × 100 mm). Positive ion mode is mobile phase A: 0.1% formic acid aqueous solution and mobile phase B: 0.1% formic acid acetonitrile. Negative ion mode is mobile phase A: 0.1% formic acid aqueous solution and mobile phase B: 0.1% formic acid acetonitrile. The injection volume is 1 μL. Data processing was performed following the method of Li [21]. The raw data collected using MassLynx V4.2 were processed with Progenesis QI software (version 2.0) for peak extraction and alignment. Metabolite identification was carried out using Progenesis QI software by connecting to the METLIN database, public databases, and Bemak’s own libraries, along with theoretical fragmentation identification. The deviation of parent ion mass was limited to 100 ppm, and that of fragment mass was limited to 50 ppm. Bioinformatics analyses of metabolites were performed using BMKCloud to assess the sample quality and control the reproducibility within the group through principal component analysis and Spearman correlation analysis. Information on the classification and pathways of identified compounds was retrieved from KEGG (Kyoto Encyclopedia of Genes and Genomes), HMDB (Human Metabolome Database), and LipidMaps (Lipid Metabolites and Pathways Database). Based on the grouping information, the multiplicity of differences was calculated, and the *p*-value for significant differences of each compound was determined using a *t*-test. OPLS-DA modeling was performed using the R language package ropls (ropls v1.16.1). Differential metabolites were screened based on the combinatorial multiplicity of the differential metabolites and the significance of the projection (VIP) value in the OPLS-DA model, with screening criteria of *p* < 0.05 and VIP > 1. The significant enrichment of differential metabolites in the KEGG pathway was assessed by using the hypergeometric distribution test [22].

### 2.5. Data Analysis

Before the statistical analyses, the data were checked for normality and variance homogeneity using SPSS (v26.0) and Levene’s test. A one-way analysis of variance (ANOVA) was performed using SPSS software, and when differences were significant, multiple comparisons were conducted using the Duncan method, with *p* < 0.05 as the criterion for significance. The relationship between microorganisms and metabolites in oat and vetch silage mixtures was analyzed using Spearman’s correlation coefficient, with *p* < 0.05 and R > 0.60 as the criteria for significant correlation.

## 3. Results

### 3.1. Silage Fermentation Quality

The fermentation characteristics of mixed oats and vetch silage for 90 days were analyzed, and the results are presented in Table 1. The levels of pH, LA, and AA in the silage were 4.12, 1.12%, and 0.22%, respectively. Both PA and BA were not detected in the silage and are, therefore, not listed. The level of AN was 16.86% TN.

### 3.2. Chemical Composition of Mixed Silage at 0 Days and 90 Days

Oat and vetch silage mixtures were tested for DM, CP, WSC, NDF, ADF, ash, and EE at 0 days and 90 days, and the results are shown in Table 2. The contents of DM, CP, WSC, NDF, ADF, Ash, and EE in CK0d were 37.74%, 13.85%, 13.00%, 50.09%, 30.40%, 6.72%, and 2.76%, respectively. The levels of DM, CP, WSC, NDF, ADF, Ash, and EE in CK90d were 34.41%, 10.58%, 8.77%, 45.08%, 24.97%, 11.05%, and 3.57%, respectively. After 90 days of silage, the contents of DM, CP, WSC, NDF, and ADF decreased, while the contents of ash and EE increased.

### 3.3. Bacterial Community Composition and Diversity of Mixed Silage at 0 Days and 90 Days

Bacterial diversity and community composition were analyzed by sequencing 16S rRNA amplicons from 0 days and 90 days for the oat and vetch silage mixtures. A total of 167,674 sequences were obtained from the 12 samples of both treatment groups, with 162,691 valid sequences remaining after the removal of invalid sequences (Appendix A). The alpha diversity indices (ACE was not significant; Chao1, Simpson, and Shannon) were significantly lower (*p* < 0.05) in the CK0d group than in the CK90d group (Figure 1A). The beta diversity of mixed feed bacteria was analyzed using NMDS based on unweighted and weighted UniFrac distances (Figure 1B,C). The results showed significant bacterial community segregation between the 0-day and 90-day mixed silage. Further identified were 17 Phyla, 24 Classes, 51 Orders, 94 Families, 182 Genera, and 254 Species of bacterial communities (Appendix A). At the phylum level, the predominant bacterial phylum in CK0d was unclassified bacteria, while the predominant bacterial phyla in CK90d were Firmicutes and Proteobacteria (Figure 1D). At the genus level, the predominant bacterial genus in CK0d was unclassified bacteria, while the predominant bacterial genera in CK90d were *Levilactobacillus* and *Pediococcus* (Figure 1E). LEfSe analysis revealed significant differences (LDA score > 4) in 13 and 20 biomarkers between the CK0d and CK90d groups, respectively (Appendix A).

### 3.4. Fungal Community Composition and Diversity of Mixed Silage at 0 Days and 90 Days

Fungal diversity and community composition were analyzed by sequencing 16S rRNA amplicons from the 0-day and 90-day oat and vetch silage mixtures. A total of 148,358 sequences were obtained from the 12 samples of both treatment groups, with 143,169 valid sequences remaining after the removal of invalid sequences (Appendix A). The alpha diversity indices (ACE was not significant; Chao1 was not significant, Simpson and Shannon) were significantly lower (*p* < 0.05) in the CK90d group than in the CK0d group (Figure 2A). The beta diversity of the mixed-feed fungi was analyzed using PLS-DA (Figure 2B). The results showed significant fungal community segregation between the 0-day and 90-day mixed silage. Further identified were 8 Phyla, 27 Classes, 52 Orders, 92 Families, 134 Genera, and 167 Species of fungi (Appendix A). At the phylum level, the predominant fungal phylum in CK0d was Basidiomycota, while the predominant fungal phylum in CK90d was Ascomycota (Figure 2C). At the genus level, the predominant fungal genera in CK0d were *Filobasidium* and *Aspergillus*, while the predominant fungal genera in CK90d were *Kazachstania* and *Pichia* (Figure 2D). LEfSe analysis revealed significant differences (LDA score > 4) in 38 and 6 biomarkers between the CK0d and CK90d groups, respectively (Appendix A).

### 3.5. Metabolite Composition of Mixed Silage at 0 Days and 90 Days

To thoroughly investigate the metabolic differences between 0-day and 90-day silage mixes, we conducted metabolomic analyses of oats and vetch. As shown by the OPLS-DA model, the R2Y for the between-group comparison was close to one (Figure 3A), and the slope of the fitted regression line for Q2Y was positive (Figure 3B), indicating that the constructed model was stable, reliable, and suitable for comparing the differences between the two groups. The PCA clustering plot revealed a distinct grouping of samples from CK0d and CK90d (Appendix A). A total of 5558 metabolites were identified in both sample groups (Appendix A). A total of 2989 differential metabolites were identified between CK0d and CK90d using *p* < 0.05 and VIP > 1 as the screening criteria (Figure 3C). Among them, 1915 metabolites were upregulated and 1074 were downregulated after silage, with the primary differential metabolites being carboxylic acids and derivatives, fatty acyls, organooxygen compounds, steroids, steroid derivatives, and flavonoids (Figure 3D,E).

An enrichment analysis of the KEGG pathway revealed that the key differential metabolic pathways induced by the differential metabolites included the biosynthesis of unsaturated fatty acids; ascorbate and aldarate metabolism; pentose and glucuronate interconversions; alpha-linolenic acid metabolism; fatty acid degradation; linoleic acid metabolism; and starch and sucrose metabolism (Figure 4).

### 3.6. Combined Analysis of Microorganisms and Metabolites

Spearman’s correlation coefficient analysis was used to examine the correlation between bacteria and fungi at the genus level (*p* < 0.05, R > 0.60). The results indicated that *Levilactobacillus*, *Latilactobacillus*, and *Pediococcus* were significantly positively correlated with *Kazachstania* and significantly negatively correlated with *Filobasidium* and *Cladosporium* (Figure 5A). In the Spearman’s correlation coefficient analysis of metabolites with bacteria, *Levilactobacillus*, *Latilactobacillus*, and *Pediococcus* were significantly positively correlated with l-methionine, l-glutamine, arachidonic acid, and linolenic acid and significantly negatively correlated with l-leucine, l-arginine, l-asparagine, and glyceric acid (Figure 5B). Further analysis using the Spearman’s correlation coefficient model of metabolites with fungi revealed that *Kazachstania* was significantly positively correlated with l-methionine, l-glutamine, arachidonic acid, and linolenic acid, whereas it was significantly negatively correlated with l-leucine, l-arginine, l-asparagine, and glyceric acid. In contrast, *Filobasidium* and *Cladosporium* showed a significant positive correlation with l-leucine, l-arginine, l-asparagine, and glyceric acid and a significant negative correlation with l-methionine, l-glutamine, arachidonic acid, and linolenic acid (Figure 5C).

## 4. Discussion

Silage is an effective method for increasing the availability of feed for ruminants [23]. The chemical composition of the forage is effectively preserved through anaerobic fermentation. In this study, we examined the changes in microorganisms, metabolites, and chemical compositions of oats and vetch mixed with fresh raw materials and subjected to anaerobic fermentation in alpine pastures, using various physicochemical analyses, high-throughput sequencing, and gas chromatography–mass spectrometry (GC-MS) techniques. Previous studies have primarily concentrated on a single raw material and have largely been confined to the analysis of either a single microorganism (bacterial or fungal) or a specific metabolite. This study is the first to integrate bacteria, fungi, and metabolites to investigate the mixed fresh raw materials and silage of oats and vetch in alpine pastures, providing a preliminary model for future research in this field.

### 4.1. Fermentation Characteristics and Chemical Content of Mixed Feeds of Oats and Vetch

In this study, the chemical composition of fresh raw materials, along with the fermentation parameters and chemical composition of silage, were determined. Dry matter (DM) is a critical factor in silage quality, and the optimal DM content for producing high-quality silage should range between 30 and 40 percent [24]. The DM content of the fresh material in this study fell within this range, significantly reducing the likelihood of silage spoilage. Silage deterioration is typically characterized by a butyric acid content of up to 0.5% DM and an increased concentration of propionic acid (0.3–0.5% DM), accompanied by significant DM loss [25]. However, this was not observed in the present study, where the dry matter content remained between 30 and 40% DM after fermentation, and no butyric or propionic acids were detected. Butyric and propionic acids impair silage fermentation by stimulating clostridial growth, which causes nutrient losses and reduces palatability. pH is a key parameter for assessing fermentation quality, and in this study, the pH of the mixed silage was below 4.20, indicating successful fermentation [26]. The decline is attributed to lactic acid bacteria (LAB) utilizing the soluble carbohydrates (WSC) in the feed to produce lactic acid (LA) and acetic acid (AA), resulting in acidification [27]. The WSC content of high-quality silage should exceed 5% DM [28] In this study, the WSC content reached 13.00% DM, significantly surpassing this requirement. Blajman demonstrated that, for forage with low sugar content, prolonged storage can lead to the conversion of homofermentative LAB to heterofermentative LAB, resulting in the production of acetic acid. However, due to the high sugar content of the oat and vetch mixture, the likelihood of acetic acid production was lower [29]. Protein degradation is a common occurrence during fermentation, influenced by plant enzymes and microbial activity [7]. The lower crude protein (CP) content and higher ammoniacal nitrogen (NH3-N) content observed in this study indicate that protein degradation occurred during storage. The ADF and NDF contents were high in this study, and it has been shown that lactic acid bacteria primarily affect the fermentation of sugars and are less effective in the degradation of complex polysaccharides, such as cellulose, making it difficult to significantly increase the degradation rate of crude fiber [30]. Silages with low pH and high lactic acid content enhance the activity of beneficial microorganisms in the rumen of ruminants, thereby improving feed utilization. However, inadequate fiber degradation can increase the ruminal load, necessitating dietary optimization through the inclusion of other roughage sources.

### 4.2. Bacterial Composition of Mixed Oats and Vetch

Alpha diversity is used to measure the internal variability of bacterial communities. In this study, the bacterial diversity of mixed feeds was analyzed using Shannon’s index, Simpson’s index, the Chao1 index, and the ACE index. The results showed that bacterial diversity was lower before silage than after fermentation. High-throughput sequencing reveals that microbial diversity in raw materials typically declines significantly after ensiling, consistent with reported decreases in bacterial diversity in other maize silage studies [31,32]. However, studies on soybean and oilseed leaf silage have shown an increase in bacterial diversity [33]. These differences may be related to variations in storage temperature and the type of raw material. Under the conditions of this study, the average daily temperatures ranged from −6 °C to 18 °C, which is lower than the optimal temperature for lactic acid bacteria (LAB) (20–45 °C), making it difficult for LAB to dominate rapidly [34]. Consequently, the silage fermentation process failed to inhibit certain microorganisms, allowing them to survive and support the growth of other microorganisms, resulting in higher microbial diversity. The principal coordinate analysis (PCoA) plot clearly showed significant differences in bacterial communities before and after silage. This result indicates that silage had a significant effect on the bacterial community of the oat and vetch mixture in alpine pastures. The microorganisms with the highest abundance in the fresh raw materials were unclassified bacteria, which could not be further classified due to the limitations of high-throughput sequencing. In the future, they can be categorized, and their main roles and mechanisms can be explored using metagenomics techniques. The main bacterial genera after silage were *Levilactobacillus*, *Latilactobacillus*, and *Pediococcus*. *Levilactobacillus* can undergo sugar fermentation, and the high soluble sugar content of the oat and vetch mixture supported the unrestricted growth of *Levilactobacillus*. The rapid consumption of water-soluble carbohydrates (WSC) via homofermentation generates lactic acid and a small amount of acetic acid, reducing the pH to 4.12. This acidification effectively suppresses the growth of spoilage bacteria and fungi, minimizes protein degradation and toxin formation, and thereby preserves the crude protein (CP) and dry matter (DM) content of the feed. Human health risks from feed nitrite arise through a cascade: microbial reduction enables intragastric nitrosation, leading to accumulation in animal tissues and subsequent dietary exposure in humans, with nitrosamines as the ultimate mediators [35]. Additionally, excessive nitrite intake can have adverse effects on human health and may lead to diseases such as hypertension and glioma [36]. *Latilactobacillus* has the ability to synthesize the enzyme nitrite reductase, so higher levels may help reduce nitrite formation [9]. This property provides significant safety benefits in silage applications. Upon silo opening, the lactic acid produced by *Pediococcus* inhibits the resurgence of saccharomyces and molds, thereby reducing the risk of aerobic spoilage and extending shelf life. However, *Pediococcus* is intolerant to low pH [37]. As a result, *Pediococcus* is less abundant compared to other lactic acid bacteria after silage.

### 4.3. Fungal Composition of Mixed Oats and Vetch

Currently, there is an abundance of research on the changes in bacterial communities during silage fermentation, while fungal research has primarily focused on aerobic spoilage of silage [38,39,40,41]. However, the development of fungal communities during fermentation, particularly in mixed oat and vetch silage, remains an area that has not been thoroughly investigated. Therefore, in this study, fungal communities in mixed oat and vetch feeds before and after silage were analyzed using ITS amplicon sequencing. The alpha diversity of the fungal community was assessed by calculating the Shannon, Simpson, Chao1, and ACE indices. These indices reflect the diversity within the fungal community, and significant differences in fungal diversity were observed before and after silage. This difference may result from the consumption of oxygen by aerobic microorganisms during the silage process, leading to the formation of anaerobic and acidic environments that are unfavorable for fungal growth, thereby reducing fungal diversity. In this study, *Filobasidium* and *Aspergillus* were found to be the dominant genera in silage, which aligns with the results of previous studies [42,43,44]. *Aspergillus* is typically considered a producer of toxins that may compromise silage safety [45]. However, *Aspergillus* is also known for its ability to produce a wide range of enzymes, including cellulases, hemicellulases, and xylanases, which may contribute to the breakdown of plant cell wall components and are widely used in the fermentation process to enhance nutritional value [46]. *Pichia* and *Kazachstania* are the dominant genera in silage feeds. *Pichia* is commonly found in aerobically spoiled silage [47,48]. In the study by [49], *Pichia* was shown to be tolerant to a wide range of environmental stresses, particularly acidic and salt stress, which explains its high abundance in silage. Furthermore, the study by [47] demonstrated that *Kazachstania* was the main Saccharomyces species in silage exposed to air, based on denaturing gradient gel electrophoresis profiles of fungal DNA and RNA. According to [50], *Kazachstania* is the predominant *Saccharomyces* genus. An increased abundance of acid-tolerant *Saccharomyces*, such as *Kazachstania* and *Pichia*, may be linked to lactic acid utilization and the production of specific metabolites. While these metabolites can enhance feed palatability, excessive yeast activity may deplete lactic acid, resulting in pH rebound and a heightened risk of secondary fermentation.

### 4.4. Metabolite Composition of Mixed Oats and Vetch

There are two sources of metabolites produced during the silage process: those directly generated by microbial activity and those formed indirectly through the degradation and transformation of existing substances in the material. The significant separation of metabolites between fresh raw material and silage in the PCA and PLS-DA scoring plots may be attributed to changes in the metabolome caused by the action of plant enzymes during the aerobic silage phase, as well as alterations in the dynamic abundance of lactic acid bacteria (LAB) strains, which led to changes in many metabolites, particularly amino acids, carbohydrates, and fatty acids, in the mixture of oats and vetch [51]. Amino acids and their derivatives play crucial physiological roles in silage and are involved in biological processes such as protein synthesis, hormone regulation, and signal transduction [52]. In this study, it was found that the content of l-methionine increased after silage, while the levels of L-Lysine and l-arginine remained relatively unchanged. L-methionine and l-lysine can alleviate oxidative stress, which is a major contributor to many diseases in ruminants. Additionally, oxidative stress, especially during the periparturient period, impacts the immunity, performance, and metabolism of ruminants, such as dairy cows [53]. Therefore, supplementing with adequate amounts of protective amino acids (l-methionine and l-lysine) can help protect animals from disease and enhance their productivity. L-arginine plays a crucial role in the adaptation of lactic acid bacteria to acidic environments [54]. Fatty acids play a vital role in providing energy to lactic acid bacteria and sustaining their vital activities. In the present study, it was found that the relative content of arachidonic acid and linolenic acid increased. Arachidonic acid, an unsaturated fatty acid with anti-inflammatory properties, is an essential fatty acid for animals and must be obtained through their diet [55]. In addition, high levels of arachidonic acid activate *PPAR-α* more efficiently, thereby increasing intramuscular fat (IMF) [56]. Linolenic acid is associated with the deposition of amino acids and fatty acids in animal muscle. An elevated fatty acid content enhances the energy density of silage, supporting caloric maintenance and improving the performance of ruminants under low-temperature conditions.

### 4.5. Microbial and Metabolite Interactions

To explore the correlation between the main microorganisms and metabolites, a Spearman correlation analysis was performed. The results showed that the metabolites positively correlated with LAB strains were also positively correlated with *Saccharomyces*. This result suggests that there is a synergistic effect between *Lactobacillus* and *Saccharomyces* during the silage process. In addition, crop characteristics, such as dry matter content and sugar content, combined with the lactic acid resistance, osmotic resistance, and substrate utilization capabilities of lactic acid bacteria, have a decisive influence on the competitiveness of the lactic acid bacterial community during silage fermentation [57]. The dry matter and soluble sugar content of the oat and vetch mixture in this study resulted in a total abundance of *Levilactobacillus* and *Latilactobacillus* of 70% after 90 days of silage fermentation. Both LAB strains exhibited significant positive correlations with l-methionine, l-glutamine, arachidonic acid, and linolenic acid, while significant negative correlations were observed with l-leucine, l-arginine, l-asparagine, and glyceric acid. However, correlation does not imply causation and is merely speculative, based on statistical and correlation parameters. It is also important to note that the absence of significant correlations between certain bacterial taxa and metabolites does not necessarily imply a lack of relationship between them [58]. Additionally, the functional characteristics of certain species may have a significant impact on community structure and ecosystem function [59]. Regarding the correlation between fungi and metabolites, the results indicated that *Kazachstania* was significantly positively correlated with l-methionine, l-glutamine, arachidonic acid, and linolenic acid, whereas it was significantly negatively correlated with l-leucine, l-arginine, l-asparagine, and glyceric acid. Based on the positive correlation between *Kazachstania* and l-methionine, as well as the elevated l-methionine content in the mixed feeds after silage, it can be hypothesized that Kazachstania may increase the l-methionine content in silage. An increase in l-lysine content was observed in whole-strain maize silage inoculated with *Saccharomyces* cerevisiae Brucellae, but there was minimal change in the l-lysine content before and after silage in this study. These results suggest that *Saccharomyces* can produce certain amino acids during forage silage, which is consistent with the findings of a previous study on alfalfa silage [60]. Nevertheless, the correlation between LAB, *Saccharomyces,* and metabolites in oats and vetch silage mixtures provides valuable insights for the screening of target LAB and *Saccharomyces.* These LAB and *Saccharomyces* can not only be used to regulate the fermentation process of silage and produce high-quality silage but also extend the functionality of silage from enhancing the nutritional value to promoting animal health and welfare. Therefore, the screening and application of silage LAB and Saccharomyces inoculants that secrete bioactive compounds (such as amino acids, small peptides, flavoring agents, bacteriocins, antimicrobial agents, vitamins, and polysaccharides) will be a key focus of future silage research.

## 5. Conclusions

In this study, we investigated changes in the microorganisms, metabolites, and nutritional quality of oat and vetch mixtures during natural low-temperature silage in alpine pastures. The regulatory mechanisms of silage under low temperature conditions and their impacts on forage nutritional value were elucidated. In the present study, the main dominant genera of fresh raw materials were unclassified bacteria, *Filobasidium*, and *Aspergillus*, while the dominant genera after silage were *Levilactobacillus*, *Pediococcus*, *Kazachstania*, and *Pichia*. Silage can lead to 77 different microorganisms and 2989 different metabolites. These metabolites include amino acids, fatty acids, and carbohydrates and are involved in a variety of metabolic pathways, such as the biosynthesis of unsaturated fatty acids; ascorbate and aldarate metabolism; pentose and glucuronate interconversions; alpha-linolenic acid metabolism; fatty acid degradation; linoleic acid metabolism; and starch and sucrose metabolism. This study systematically elucidated the synergistic mechanisms between microorganisms and metabolites during low-temperature silage. This study also offers novel insights for the future identification of the biomarkers involved in regulating mixed silage quality in alpine pastures. Future research could further investigate the relationship between microbial functional genes and metabolic pathways, thereby providing a theoretical foundation for producing high-quality mixed silage in alpine pastures.

## Figures and Tables

**Figure 1 microorganisms-13-01535-f001:**
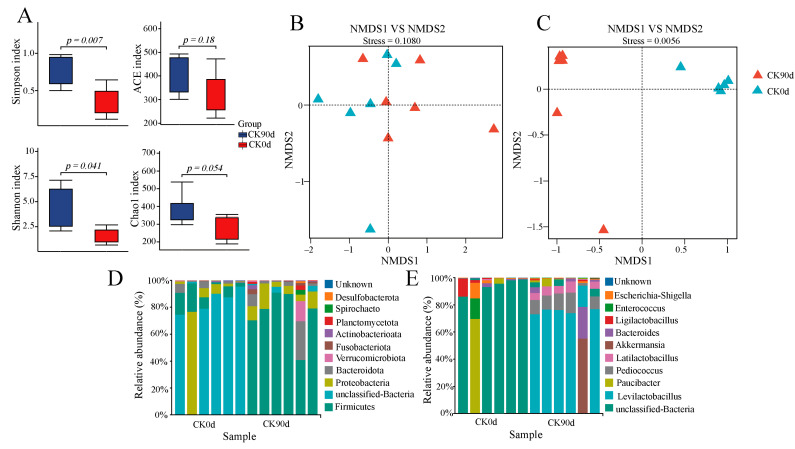
Diversity and variability of bacterial communities in a mixed oat and vetch. (**A**) Variation in alpha diversity of bacterial communities. (**B**,**C**) Differences in bacterial communities across treatments, calculated using unweighted UniFrac and weighted UniFrac distances, respectively, with coordinates calculated using principal coordinate analysis. (**D**,**E**) Relative abundance of bacterial phylum and genus.

**Figure 2 microorganisms-13-01535-f002:**
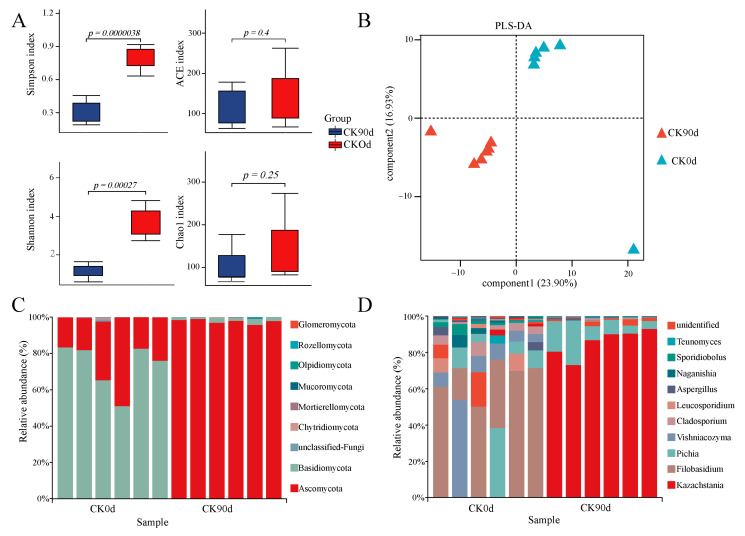
Diversity and variability of fungal communities in a mixed oat and vetch. (**A**) Variation in alpha diversity of fungal communities. (**B**) Variability of fungal communities under different treatments. (**C**,**D**) Relative abundance of fungal phyla and genera.

**Figure 3 microorganisms-13-01535-f003:**
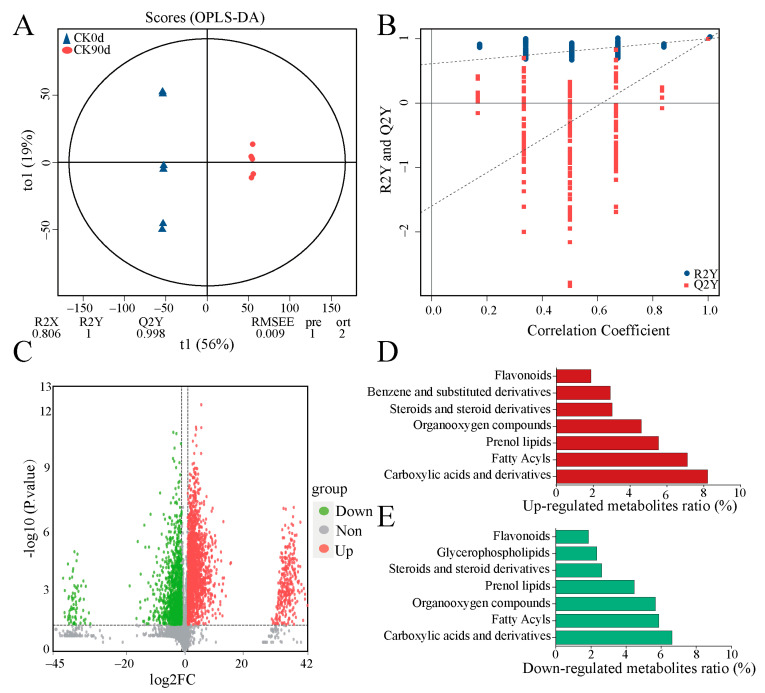
Metabolite analyses of oats and vetch mixed diets. (**A**) Plot of OPLS-DA scores for metabolites. (**B**) OPLS-DA model replacement test plot. (**C**) Volcano plot showing up- and down-regulation of metabolites. (**D**,**E**) Summary of analyses of up- and down-regulated metabolites.

**Figure 4 microorganisms-13-01535-f004:**
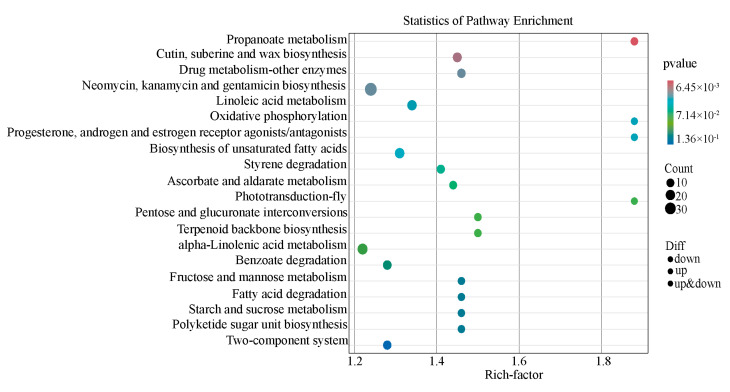
KEGG pathway enrichment analysis of different metabolites.

**Figure 5 microorganisms-13-01535-f005:**
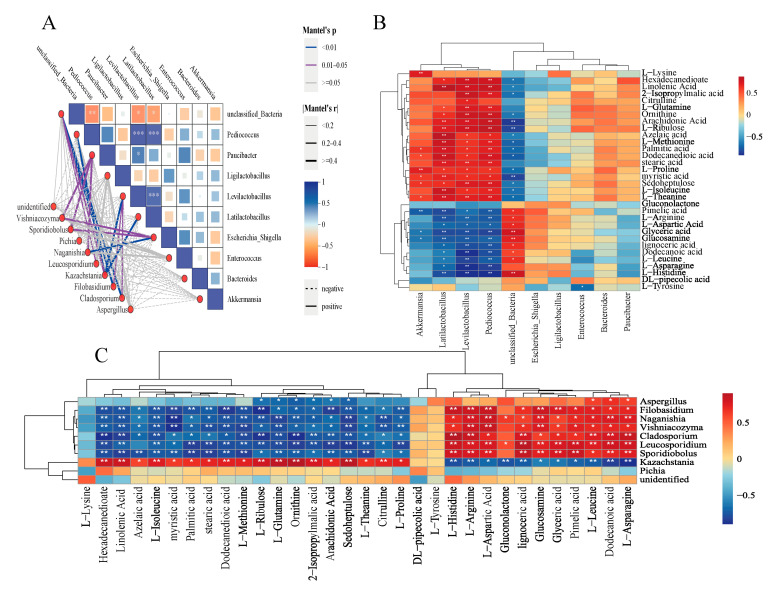
Correlation analysis between microorganisms and metabolites of mixed oat and vetch. (**A**) Heat map of bacterial–fungal networks; (**B**) Heat map of bacterial–metabolome correlations; (**C**) Heat map of fungal–metabolome correlations. *, **, and *** represent *p* < 0.05; *p* < 0.01; *p* < 0.001.

**Table 1 microorganisms-13-01535-t001:** Fermentation characteristics of mixed silage of oats and vetch.

Items	Content
pH	4.12
AN (%TN)	16.86
LA (%DM)	1.12
AA (%DM)	0.22
PA (%DM)	0.00

DM, dry matter; LA, lactic acid; AA, acetic acid; PA, propionic acid.

**Table 2 microorganisms-13-01535-t002:** Chemical composition of mixed oat and vetch silage at 0 and 90 days.

Treatment	Items						
	DM (%)	CP (% DM)	WSC (% DM)	NDF (% DM)	ADF (% DM)	Ash (% DM)	EE (% DM)
CK0d	37.74 ± 0.26 a	13.85 ± 0.13 a	13.00 ± 0.76 a	50.09 ± 0.46 a	30.40 ± 0.19 a	6.72 ± 0.12 b	2.76 ± 0.03 b
CK90d	34.41 ± 0.11 b	10.58 ± 0.10 b	8.77 ± 0.03 b	45.08 ± 0.09 b	24.97 ± 0.10 b	11.05 ± 0.11 a	3.57 ± 0.17 a

Significant differences between treatments are indicated by lower case letters, *p* < 0.05; DM, dry matter; WSC, water-soluble carbohydrate; CP, crude protein; ADF, acid detergent fiber; NDF, neutral detergent fiber; ash, crude ash; EE, ether extract.

## Data Availability

The original contributions presented in this study are included in the article/Appendix A. Further inquiries can be directed to the corresponding author.

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
