# Peer review of "Dynamics of Microorganisms and Metabolites in the Mixed Silage of Oats and Vetch in Alpine Pastures, and Their Regulatory Mechanisms Under Low Temperatures"

_microorganisms, 2025, doi:10.3390/microorganisms13071535_

Round 1

Reviewer 1 Report

Comments and Suggestions for Authors

The study describes a mixture of vetch and oats undergoing fermentation over 90 days at a temperature below 20°C which was performed as six replicates achieving pH 4.  The main physicochemical properties such as cellulose and ash showed expected significant differences.  DNA was extracted to determine the bacterial and fungal diversity which revealed an increase in bacterial diversity which was significantly different using 3 out of 4 diversity measurements and was distinctly different.  The main bacteria were Levilactobacillus, Latilactobacillus, and Pediococcus.  Fungal diversity shifted from Basidiomycetes being predominant to be replaced by Ascomycetes (Kazachstania and Pichia).  Metabolites were measured using GC revealing upregulation and downregulation.  In addition, some correlation analysis was performed between dominant microorganisms and amino acids showing some were significantly upregulated and some were downregulated.

Main critical question

There are only two samples, at the start and at the end of decomposition, with six replicates of each.  Correlation analysis was performed on this data but there are only two variables which I do not think is enough.  It should be a lot more and the more the better.

Abstract claims that amino acids were increased but instead correlations were performed between dominant microorganisms and amino acids.  This is not correct.  It should be possible to obtain quantifiable data from GC data of these amino acids to show possible changes.

In abstract  The reduction in amino acids....    This sentence sounds contradictory to the previous sentence.  Needs rewording.

Section 2.1 what is the proportion of oats and vetch?

section 3.2 and Fig 1  Make table instead and give more of a general description of results.

Section 3.5  Use of capitalization in mid-sentence is unnecessary e.g. Biosynthesis, ascorbate, Pentose at bottom of page 7.

Section 4.2 Representation of results rather than discussion of results.

References  Some of the journal titles are not in title capital letters and microorganism name not abbreviated.  Also title of articles in title capital letters.

Author Response

Dear reviewers

Reviewer 2 Report

Comments and Suggestions for Authors

Your study is very comprehensive, showing interesting results that shed new light on silage processes. Below, I list the questions that arose during the review of the document.

Materials and Methods

Lines 103-104. How often did you analyze samples?

Lines 191-197. Were the experiments performed in triplicate?

Table 1. In the table footnote, indicate BA, butyric acid.

Figure 1. What are the y-axis units?

Line 332. What advantage does the fact that butyric and propionic acids were not detected have on silage?

Lines 333-338. What could be the reason for the increase in water-soluble carbohydrates? Is it due to the raw material? If the presence of soluble sugars is beneficial, what is more advisable: extending the fermentation time or the raw material used for silage?

Lines 399-429. Are there silage processes that use the same raw materials? If so, how do their results compare with those obtained? Furthermore, is this Aspergillus present in silage really an advantage? Isn't its presence more problematic due to its potential production of mycotoxins?

Lines 430-438. If you are referring to the detection of enzymes from the raw material, isn't this due to contamination during sequencing? Ultimately, sequencing is not selective and will detect everything in the sample, including plant enzymes; this doesn't mean they participate in the silage degradation process.

Lines 441-449. At what point in the silage process were these amino acids detected? In an anaerobic process, reducing, not oxidative, conditions prevail.

Comments on the Quality of English Language

Acceptable English

Author Response

Dear reviewers

Reviewer 3 Report

Comments and Suggestions for Authors

The manuscript subject relates to the microbial biodiversity of low-temperature fermented silage correlated to metabolites’ profile, targeting the quality of the final feed. The interest of the subject is moderate from a practical perspective, however, the methodological approach is cutting-edge, which makes it interesting from a theoretical point of view.

Generally, the information provided under Introduction section is not sufficiently convincing on why such study was necessary to be conducted from a practical/applicative perspective.

Methodology is very well elaborated and explained in the experimental context.

The results are clearly presented and supported with tables and figures. However, the Supplementary Tables and Figures (see lines 224-237) were not provided/uploaded in the platform.

Generally, the discussions are well argued and supported with relevant references.

Line 365 & 367: Please provide in the text the methods used by the cited authors (Xin, et al., 2023; Mao, et al., 2022; Wang, et al., 2022) to characterize the microbial biodiversity in their study and make a comment in this regard.

Lines 388-390: please provide an explanation on how different biochemical processes in the feed/silage may affect in the end the human health as you stated. Also, please provide a reference for the statement “excessive nitrite intake can have adverse effects on human health and may lead to diseases such as hypertension and glioma”.

Line 394: replace saccharomyces with Saccharomyces (in Italic)

Line 425 & 426 : you wrote “Kazachstania is the predominant Saccharomyces genus” and “acid-tolerant Saccharomyces, such as Kazachstania and Pichia”. Please explain if both yeast Kazachstania and Pichia are Saccharomyces or non-Saccharomyces on genus level (even if they are all Saccharomycetaceae).

Author Response

Dear reviewers

Round 2

Reviewer 1 Report

Comments and Suggestions for Authors

My mistake as I thought there were only two variables.  Personally, not sure how much this adds to the manuscript and would have removed it to make the manuscript more concise.

Line 87 This was corrected but states that it was sown rather being harvested.

Line 248 This text seems to be in a different font.

Line 389 Remove journal title etc

Page 13 Words in middle of sentence which are capitalized - check the remaining document.

Line 464 Italics for microoraganisms

Author Response

Dear Comments 1
